# A Bias Drift Suppression Method Based on ICELMD and ARMA-KF for MEMS Gyros

**DOI:** 10.3390/mi14010109

**Published:** 2022-12-30

**Authors:** Lihui Feng, Le Du, Junqiang Guo, Jianmin Cui, Jihua Lu, Zhengqiang Zhu, Lijuan Wang

**Affiliations:** 1School of Optics and Photonics, Beijing Institute of Technology, Beijing 100081, China; 2Key Laboratory of Photonics Information Technology, Ministry of Industry and Information, Beijing 100081, China; 3School of Integrated Circuits and Electronics, Beijing Institution of Technology, Beijing 100081, China; 4Science and Technology on Communication Networks Laboratory, Shijiazhuang 050050, China; 5Beijing Institute of Aerospace Control Devices, Beijing 100039, China

**Keywords:** micro-electro-mechanical-system gyros, interpolated complementary ensemble local mean decomposition, autogressive moving average, Kalman filtering

## Abstract

The applications of Micro-Electro-Mechanical-System (MEMS) gyros in inertial navigation system is gradually increasing. However, the random drift of gyro deteriorates the system performance which restricting the applications of high precision. We propose a bias drift compensation model based on two-fold Interpolated Complementary Ensemble Local Mean Decomposition (ICELMD) and autoregressive moving average-Kalman filtering (ARMA-KF). We modify CELMD into ICELMD, which is less complicated and overcomes the endpoint effect. Further, the ICELMD is combined with ARMA-KF to separate and simplify the preprocessed signal, resulting improved denoising performance. In the model, the abnormal noise is removed in preprocess by 2σ criterion with ICELMD. Then, continuous mean square error (CMSE) and Permutation Entropy (PE) are both applied to categorize the preprocessed signal into noise, mixed and useful components. After abandon the noise components and denoise the mixed components by ARMA-KF, we rebuild the noise suppression signal of MEMS gyro. Experiments are carried out to validate the proposed algorithm. The angle random walk of gyro decreases from 2.4156${}^{\circ }$/$\sqrt{h}$ to 0.0487${}^{\circ }$/$\sqrt{h}$, the zero bias instability lowered from 0.3753${}^{\circ }$/*h* to 0.0509${}^{\circ }$/*h*. Further, the standard deviation and the variance are greatly reduced, indicating that the proposed method has better suppression effect, stability and adaptability.

## 1. Introduction

Micro-Electro-Mechanical-System (MEMS) gyro is used as inertial sensor to measure angular rates in automatic control, wearable devices, industrial, consumer electronics, inertial guidance systems and other fields. With its advantages in small size, low power consumption, integration, light weight and mass production, MEMS gyro has been widely applied [1,2,3]. Limited by both the principle and mechanical structure, the performance of MEMS gyro is worse than some kinds of gyros such as fiber optic and laser gyros. It is easily affected by the imperfection of the structure and environmental factors, which limit its precision and further applications [4,5]. Therefore, methods of optimizing the structure and signal denoising are essential to improve the performances of MEMS gyros.There are about three kinds of methods to improve the performance of MEMS gyro [6]:Physical methods:These methods generally include optimizing the structure and improving the industrial technology, which are completed in the design and production stages. The commonly used methods are laser quality adjustment [7], adding mass block [8] and circuit compensation;Linear or single modelsThese are usually signal analyzing or filtering methods, such as modeling of time series [9], Kalman filtering (KF) (including modified KFs) [10,11,12] and wavelet threshold filtering [13], etc. In [11], Yi proposed a robust Kalman filtering under model uncertainly. Qi, W brought out time-varying Kalman Filters for advanced manufacturing [12]. Their work have widened the applications of KF. Limited by the technology and objective factors, linear or single models have certain bottlenecks.Mixed models:Including several algorithms for signal analyzing and denoising in one model, which have been widely applied nowadays.

Recently, several adaptive signal analyzing methods have been studied and applied for MEMS gyros [14,15,16,17,18,19], i.e., Empirical Mode Decomposition (EMD) and Local Mean Decomposition (LMD). EMD decomposes the signal into components called intrinsic mode function (IMF), while LMD decomposes the signal into components called product function (PF). However, the EMD and LMD have the disadvantages of modal aliasing and endpoint effect. Then, Wu proposed an integrated empirical mode (EEMD) algorithm to better suppress the modal aliasing by adding white noise [20]. However, the EEMD has the shortcomings of not completely eliminating white noise and error during the processes of signal decomposition and reconstruction. Further, Yeh et al. proposed a complementary integrated empirical mode decomposition (CEEMD) method by adding auxiliary white noise in the form of positive and negative pairs to eliminate the residual white noise in the reconstructed signal and reduce the impact of modal aliasing [21]. The principle of ELMD and CELMD is same as EEMD and CEEMD. In [19], EMD has been applied to decompose the output signal for the hard-threshold denoising of MEMS gyro. However, this method does not classify the IMFs, which diminish the role of EMD, and the hard-threshold cannot filter the signal well. Then, a noise reduction algorithm based on EMD and wavelet threshold was proposed for MEMS accelerometer in [22]. This method classifies the IMFs into two parts, and applies wavelet threshold to denoise which behaves better than hard-threshold. Guo proposed hybrid methods for the noise reduction over MEMS gyro signal [23], two indexes divide the IMFs into three parts, soft interval thresholding, soft thresholding, or forward-backward linear prediction were selected to reduce the noise contained in the mixed IMFs. Compared with EMD, LMD can reflect all useful characteristics of the original signal, and the decomposition result is more accurate [24]. Li applied LMD and parabolic tracking time-frequency peak filtering (PTTFPF) to reduce noise [25]. Over all, the EMD or LMD were applied for denoising the signal of MEMS gyro, resulting in the decomposed components to be classified. The methods mentioned above is shown in Table 1.

**Table 1 micromachines-14-00109-t001:** Comparison of denoising effects.

Denoising Methods	Classification	Filtering
EMD-thresholding [19]	/	Hard-threshold
EMD-wavelet threshold [22]	CMSE	Wavelet threshold
EMD- thresholding [23]	CMSE and $l_{2}$-norm	Soft interval thresholding, soft thresholding, or forward-backward linear prediction
LMD- parabolic tracking time-frequency peak filtering (PTTFPF) [25]	Sample entropy (SE)	Frequency peak filtering (PTTFPF)
Proposed method	CMSE and PE	ARMA-KF

In view of the advantages of CELMD method in signal adaptive decomposition [26,27], we consider its applications in signal denoising for MEMS gyro. We propose a MEMS gyro suppression model based on ICELMD and ARMA-KF. To reduce the affect of abnormal noise that some methods may ignore, we decompose the gyro bias drift signal by ICELMD and eliminate abnormal noise components by the 2σ criterion as preprocess. Different from [22,23], we exploit the continuous mean square error (CMSE) and Permutation Entropy (PE) to classify PF into three parts as noise components, mixed components and useful components which consider both the energy and complexity of PFs. Then, we use autogressive moving average-Kalman filtering (ARMA-KF) to denoise the mixed components which is suitable for gyro signal and easy to implement. Finally, the signal is rebuilt by abandoning the noise components, combining the useful components and denoised mixed components. We clearly state and summarize the main novelties of this paper as follows:Proposing a novel signal decomposition method, called Interpolated Complementary Ensemble Local Mean Decomposition (CELMD), shorted as called Interpolated CELMD or ICELMD, which cut the required time cost of signal decomposition and improve the endpoint effect.Applying ICELMD to analyze and category the MEMS gyros’ signal, and combine the autogressive moving average-Kalman filtering (ARMA-KF) to denoise the preprocessed signal.

According to the experiments, compared with the original signal, the methods improve the standard deviation by about 86.82%, and reduce the allan variance about 90%.

The rest of this paper is organized as follows. Section 2 describes the ICELMD and the improvements. Section 3 introduces the method of classification and denoising. Section 4 shows the experimental verification. Finally, conclusions are given.

## 2. Interpolated CELMD

### 2.1. Interpolated LMD

The LMD is an adaptive way to analyze signal, but it still has many problems limited its applications. To make better use of LMD, we proposed ILMD, the flow chart of ILMD is shown in Figure 1. The principle of LMD is in Appendix A.

The LMD is more complex than EMD. In terms of the process steps, LMD is a triple iterative process, including the sliding average, the generation of each PF component, and the decomposition. The LMD involves a sliding average process, in which the local mean and the envelope estimation curves of the signal need to repeat the sliding average operation. Therefore the generation of each PF component undergoes several iterations. To solve this problem, we apply the linear interpolation into LMD, which is as follows:Perform linear interpolation on the maximum and minimum points of the signal x(t) to obtain the corresponding upper and lower envelopes as A(t) and B(t).Calculate the local mean curve $m_{11}\left(t\right)$ and the envelope estimation function $a_{11}\left(t\right)$ according to the upper and lower envelopes:
(1)$$\left\{\begin{array}{cc} m_{11}\left(t\right)=\frac{A(t)+B(t)}{2} & \\ a_{11}\left(t\right)=\frac{\left|A(t)-B(t)\right|}{2} \end{array}\right.$$

The above steps of 1. and 2. replace the step 1. of LMD in Appendix A; therefore, the improved LMD has only two iterations, which significantly improves the decomposition efficiency. The linear interpolation can retain more local signal than the sliding average. We call this as Interpolated LMD (ILMD).

Meanwhile, there are three types of components in the PFs of the preprocessed signal. The noise components are discarded, the mixed components are filtered, and the useful components are retained. For LMD, the time spent by different orders of PF is not much different, so cutting the number of PFs can also greatly reduce the complexity of decomposition. In summary, when there are multiple PFs, the time required for the procedure can be reduced by decreasing the number of PFs. For specific implementation, the gyro data can be decomposed and classified to obtain a suitable empirical value. The empirical value of PF number used for the gyro in our experiment is 11, which means, after the signal is decomposed, there are 10 PFs and 1 residual signal.

We know from the decomposition steps of LMD that it is necessary to obtain the position of the extreme value points of the signal and operate on the local extreme value. However, for the processed finite-length signal, the extreme points at both ends are not well defined while fluctuating frequently, resulting in small distances and close distributions between the extreme points. The LMD method directly considers endpoints as extreme points, which is not consistent with the actual trend of the MEMS gyro signal; this inaccurate boundary judgment can cause distortion. As the iteration proceeds, the distorted extreme point will affect the internal data of the signal, and finally cause the decomposition result to be distorted, which is the endpoint effect. To alleviate the effect of endpoint, it is necessary to accurately confirm the extreme point condition at the signal endpoint. We applied a boundary processing method to solve this problem. Taking the left boundary as an example, the first maximum value of the left boundary is L${}_{max}$, the first minimum is L${}_{min}$, and the left endpoint value is L. The method applied in step 1. of LMD in Appendix A to find the right endpoint. The specific method is as follows:If the first maximum value of the left boundary appears earlier than the minimum value, and the left endpoint value L is greater than the first minimum value, then the left maximum value is L${}_{max}$, and the left minimum value is L${}_{min}$, otherwise the left maximum value is L${}_{max}$, and the left minimum value is L.If the first maximum value of the left boundary appears later than the minimum value, and the left endpoint value L is greater than the first maximum value, then the left maximum value is L, and the left minimum value is L${}_{min}$, otherwise the left maximum value is L${}_{max}$, and the left minimum value is L${}_{min}$.

### 2.2. Interpolated CELMD

According to the aforementioned methods, the ILMD is less complicated and more accurate than LMD, but its performance still needs to be improved. Combining the ILMD and the complementary white noise, we obtain the proposed interpolated CELMD (ICELMD). The modification is achieved by using white noise with zero-mean characteristics. By adding Gaussian white noise to the original signal, then the decomposed signal with white noise by ILMD is obtained. Repeating the aforementioned ILMD several times, adds different white noise to the original signal and calculates the average of all the decomposed PF components to obtain the final decomposition result. However, when the white noise is not large enough, it may lead to a bad decomposed result. More white noise means more time to calculate over a smaller signal to noise ratio (SNR). To make better use of the white, complementary white noise are added to the original signal as auxiliary noise to eliminate the residual noise. Meanwhile, the calculation time and SNR can be reduced. The flow chart of original signal into PFs by ICELMD is shown in Figure 2. Observing from Figure 2, m pairs of complementary Gaussian white noise are added into the original signal. Then, perform ILMD on every noise-containing signal $x_{i}\left(t\right)$, and obtain 2m PF groups. Combine all 2m *i*-order $PF_{\left(i,k\right)}$ of PF groups, divided by the number of groups 2m to obtain the real *i*-order $PF_{i}$.

## 3. Proposed Denoising Method with Classification

The overall structure of proposed method is shown in Figure 3, which is divided into two parts: preprocess and denoise. In the preprocess part, the original signal is decomposed by ICELMD, then 2σ criterion is applied to remove abnormal noise in PFs. In the denoise part, the preprocessed signal is decomposed by ICELMD, then CMSE and PE are exploited to classify the PFs into three parts, noise, mixed and useful components. Abandoning the noise components, denoising mixed components, keeping the useful components, then we obtain the denoised signal.

### 3.1. Preprocessing

Before classifying the PF components, it is necessary to preprocess the gyro signal. Eliminating the abnormal noise in the output of gyro is the aim of the preprocess. Due to the changes of the external environment or the influence of vibration and shock, abnormal noise with large amplitude is generated, resulting the distortion of output signal. The frequency of the abnormal noise signal is generally different from random drift noise and real signal. Therefore, after ICELMD, the abnormal noise is generally decomposed into several PF components to be identified and analyzed. By preprocessing, the influence of abnormal noise is reduced. Specifically, applying the 2σ criterion of the limited error, the abnormal noise components could be identified. Assume the original signal be Y(t), set the standard deviation as σ, find the max vaule (peak) of each PF component as $A_{i}$. According to the 2σ criterion, remove the components with peak large than 2σ. Recombining the rest PF components, we obtain the recovered X(t).

### 3.2. Denoising with Classification

#### 3.2.1. Classification

After preprocessing, the PF components are classified before being denoised. First, X(t) is decomposed into PF components of a fixed number L by ICELMD. Then the components are classified into the components of noise, mixed, and useful signals. The classification needs the parameters of $M_{1}$ and $M_{2}$. The components order $n\leq M_{1}$ are the noise components, the order $M_{1}<n\leq M_{2}$ are the mixed components, and the order $n>M_{2}$ are the useful components. During classification, the parameter of $M_{1}$ is obtained by calculating the CMSE of each component, and the specific steps are as follows [22]:Incrementally select the PFs for reconstruction
(2)$$z_{k}=\sum_{i=k}^{L}PF^{i}\left(t\right)+residual\left(t\right)\;k=1,2\cdots L$$Calculate the Euclidean distance between adjacent reconstructed signals
(3)$$CMSE\left(z_{k},z_{k+1}\right)=1/N\sum_{i=1}^{N}{\left(z_{k+1}\left(t_{i}\right)-z_{k}\left(t_{i}\right)\right)}^{2}\;k=1,2\cdots L-1$$$M_{1}$ = argmin[CMSE(z${}_{k}$,z${}_{k+1}$)] 1≤k≤L−3The range of *k* is limited to prevent $M_{1}$ from being too large

The CMSE represents the energy of the current PF component. Generally speaking, the energy in the signal is mainly concentrated in the useful signal, while the energy of the noise is very small due to its disorder and randomness. Compared with other types of components, the noise component has the smallest energy, so the minimum CMSE must appear in the last noise PF component, and find the order of PF with the minimum CMSE value as $M_{1}$. The components of order 1 to $M_{1}$ are noise components.

The parameter $M_{2}$ is used to distinguish the mixed components from the useful components by PE. Here, PE examines the PF components and assigns a non-negative number to each sequence, with large values correspond to more complexity or irregularity, which is calculated as follows:Reconfiguration in Phase Space for PFsA set of time series X of length N is reconstructed in phase space to obtain the matrix Y as
$$\left[\begin{array}{cccc} x(1) & x(1+t) & \cdots & x(1+(m-1)t)\\ x(2) & x(2+t) & \cdots & x(2+(m-1)t)\\ x(j) & x(j+t) & \cdots & x(j+(m-1)t)\\ \vdots & \vdots & \vdots & \vdots \\ x(K) & x(K+t) & \cdots & x(K+(m-1)t) \end{array}\right]$$
where *m* is the embedded dimension, *t* is the delay time, and *K* = *N*− (*m*− 1)*t*. Each row in the matrix *Y* is a reconstructed component, and there are altogether K reconstructed components.Rearrange the altogether reconstructed componentArrange each reconstructed component in ascending order to obtain the column index of each element position in the vector to form a set of symbolic sequences.
(4)$$S\left(l\right)=j_{1},j_{2},\cdots ,j_{m},l=1,2,\cdots ,k\;k\leq m!\;j_{n}=x\left(j+\left(i_{n}-1\right)t\right)$$The m-dimensional phase space maps different sequences of symbols with a total of m!Calculate the probabilityCalculate the number of each symbol sequence divided by the total number of different symbol sequences as the probability of that symbol sequence
(5)$$P_{1},P_{2},\cdots ,P_{k}$$Calculate PE and normalization
(6)$$H_{pe}=-\sum_{j=1}^{k}P_{j}ln\left(P_{j}\right)$$
(7)$$0\leq H_{pe}=H_{pe}/ln\left(m!\right)\leq 1$$

#### 3.2.2. Filtering

After confirming $M_{1}$ and $M_{2}$, it is necessary to filter the mixed components before reconstruction to improve the performance of MEMS gyro. First, the PF orders of $M_{1}$ to $M_{2}$ are reconstructed, denoted as G(t), G(t) is properly modeled and filtered. Calculate the autocorrelation and partial autocorrelation functions of the sequence, and choose the proper model of time series according to the Table 2.

After the model is confirmed, use the criteria of Akaike and Bayesian to determine the order of the time series model. The parameters of each order are calculated according to the least squares estimation. Taking the second-fourth-order ARMA model as an example, the model (signal) could be expressed as:(8)$$G_{t}=\beta_{1}G_{t-1}+\beta_{2}G_{t-2}+\epsilon_{t}+\alpha_{1}\epsilon_{t-1}+\alpha_{2}\epsilon_{t-2}=\chi G_{t-1}+\delta \epsilon_{t}$$

KF is an effective method to denoise the time series; the state and observation equations are as follows:(9)$$\left\{\begin{array}{cc} X_{k}=\Phi_{k-1}X_{k-1}+\Gamma_{k-1}W_{k-1} & \\ Z_{k}=H_{k}X_{k}+V_{k} \end{array}\right.$$
where the $X_{k}$ is the state of the system; $Z_{k}$ is the observations; $\Phi_{k-1}$ and $\Gamma_{k-1}$ is the system state-transition matrix; H${}_{k}$ is the measurement matrix; $W_{k-1}$ is the system noise; $V_{k}$ is the measurement noise. $W_{k-1}$ and $V_{k}$ are mutually independent white noise sequences. The statistical characteristics are as follows [10]:(10)$$\begin{array}{c} W_{k}∼\left(0,Q_{k}\right)\;V_{k}∼\left(0,R_{k}\right)\\ E\left[W_{k}\right]=E\left[V_{k}\right]=0\;E\left[W_{k}V_{j}^{T}\right]=0\\ E\left[V_{k}V_{j}^{T}\right]=R_{k}\delta_{kj}\;E\left[W_{k}W_{j}^{T}\right]=Q_{k}\delta_{kj} \end{array}$$
where $R_{k}$ is the covariance matrix of measurement noise; $Q_{k}$ is the covariance matrix of process noise; $\delta_{kj}$ is the Kronecker delta function. The estimate of the observation $\hat{X_{k}}$ at time k can be solved by the following:

Error covariance extrapolation:(11)$$P_{k}^{-}=\Phi_{k-1}P_{k-1}^{+}\Phi_{k-1}^{T}+Q_{k-1}$$

Kalman gain matrix
(12)$$K_{k}=P_{k}^{-}H_{k}^{T}{\left(H_{k}P_{k}^{-}H_{k}^{T}+R_{k}\right)}^{-1}$$

State estimate extrapolation
(13)$${\hat{x}}_{k}^{-}=\Phi_{k-1}{\hat{x}}_{k-1}^{+}$$

State estimate observational update
(14)$${\hat{x}}_{k}^{+}={\hat{x}}_{k}^{-}+K_{k}\left(Z_{k}-H_{k}{\hat{x}}_{k-1}^{-}\right)$$

Error covariance update
(15)$$P_{k}^{+}=\left(I-K_{k}H_{k}\right)P_{k}^{-}$$

$P_{k}$ denotes the covariance of the estimation error; *k* means the time step *k*; the “+” superscript denotes that the estimate is a posteriori; while the “−” superscript denotes that the estimate is a priori. The “ $\hat{x}$ ” denotes the estimate of “*x*”. $x^{\prime }\left(t\right)$ denotes the final denoised signal. The $R_{k}$ and $Q_{k}$ could be determined in the ARMA model. Based on the ARMA model, KF is performed to mixed components. After KF, the denoise signal is assumed to be $G^{\prime }$(t), then the zero bias suppression signal is rebuilt as:(16)$$x^{\prime }\left(t\right)=\sum_{M_{2}+1}^{L}PF^{i}\left(t\right)+G^{\prime }\left(t\right)+residual\left(t\right)$$

## 4. Experiment

Two types of signals are denoised to verify the effectiveness of proposed method. The equipment used in the experiment is shown in Figure 4, which includes:Temperature-controlled oven is used to control temperature and provide a turntable.PC, with labview program is exploited to collect gyro angular velocity.Temperature-controlled single-axis rate turntable, capable of controlling the single-axis turntable in the oven, with optional position mode, rate mode, etc.Controllable power supply for MEMS gyro.Evaluation circuits, package and transmit multiple gyro output signals together.MEMS gyros, measuring angular velocity.

The controller of single-axis rate turntable is near the PC, and the turntable is in the temperature-controlled oven. The MEMS gyros are mounted on a turntable by a fixed table. The Evaluation circuits connecting the MEMS gyros to PC. In the experiment, we use the temperature oven to control the temperature, the turntable to control the input velocity. To evaluate the suppression effect of the proposed denoising method, CEEMD and LMD are compared during experiment of data processing. Use different methods to decompose, then, applying the same method of ARMA-KF to denoise.

### 4.1. Denoising Processing

The processing steps are listed as:Put the gyro in the temperature-controlled oven, keep the temperature at 25 °C for two hours to ensure the stability of the gyro.Keep angular velocity input as zero for two hours and the sampling rate as 1 Hz as static state test of MEMS gyro;After 2 h, stop collecting data, obtain the raw data of MEMS gyro.

The raw data are collected as the output of the gyro shown in Figure 5. After collecting data, i.e., the raw data are acquired during the static state. The ICELMD is performed over the original signal to obtain the decomposed PF and residual signals. Then, we calculate the variance of these components and original signal. By the 2σ criterion, the components with excessive variance are eliminated. Then, the preprocessed signal is decomposed by ICELMD, whose result is shown in Figure 6. After decomposition, 10 PF components and a residual signal are obtained, where the first and the last PFs correspond to the original signal and the residual signal, respectively. The CMSE and PE of each component are shown in Figure 7. It can be observed that the CMSE gradually decreases with the increasing of the PF order at the beginning, and the CMSE value of the 5th-order component is greater than that of the 3rd-order component. Therefore, we conduce that the first three orders are dominated by noise components, which means that the first parameter M${}_{1}$ is 3. It can be observed that PE of order 10 is less than that of order 9, so the second parameter M${}_{1}$ is 9. Then, we call the last PF and residual signal as useful components. From the result of classification, we need to denoising the signal as Equation (Equation 16). Remove noise components, use ARMA to model PF components from order 4 to 9, denoise these components by KF, and remain a useful component.

Figure 8 shows the denoising result; for the static state, probably the abnormal noise with excessive variance do not exist, so the preprocessed signal is the same as the original signal. However, it may exist in some low accuracy MEMS gyro; to show the full decomposition effect, we also use the gyro angular velocity data acquired by MPU6050 in zero input state, whose denoising result is shown in Figure 9. It can be observed from both Figure 8 and Figure 9 that our methods have significant improvement on the signal optimization and noise reducing.

To better show the denoise effect of the proposed model, we make more experiments with several gyros and MPU6050, the denoised results are shown below. Figure 10 shows different denoising results of ICELMD, Figure 11 shows the Allan variance of different denoising results of ICELMD. Observing that, we can see that our model has performed well in repeated experiments.

### 4.2. Comparisons

Figure 12 shows the denoising results of ICELMD, LMD and CEEMD, which are obvious and effective. The denoising gap between CEEMD and LMD is subtle. However, the denoising effect of ICELMD is greater than that of CEEMD and LMD. Because the random drift in signal of MEMS gyro has the characteristics of slow time-varying and non-smooth, the Allan variance is applied to verify the performance of different methods. Allan variance of three methods are shown in Figure 13. Among which, the ICELMD is much better than the others, and the parameters of Allan variance of different methods are shown in Table 3. Observing from Table 3 that compared with the original signal, the angle random walk decreased about 97.98%, and other parameters also perform well. Compared with LMD and CEEMD, the angle random walk decreased from 0.1585${}^{\circ }$/$\sqrt{h}$ and 0.6262${}^{\circ }$/$\sqrt{h}$ to 0.0487${}^{\circ }$/$\sqrt{h}$, and the zero bias instability lowered from 0.1642${}^{\circ }$/*h* and 0.3147${}^{\circ }$/*h* to 0.0509${}^{\circ }$/*h*. Moreover, compared with [25,28], the standard deviation suppression ratio improves from 57.87%, 67.22% to 86.82%, and the variance suppression ratio increases from 82.25%, 89.26% to 98.29%, respectively.

**Table 3 micromachines-14-00109-t003:** Comparison of denoising effects.

Denoising Methods	Angle Random Walk (N) (${}^{\circ }$/$\sqrt{h}$)	Zero Bias Instability (B) (${}^{\circ }$/*h*)	Standard Deviation Suppression Ratio	Variance Suppression Ratio
Original signal	2.4156	0.3753	-	-
ILMD-PTTFPF [25]	-	-	57.87%	82.25%
RNN-UKF [28]	-	-	67.22%	89.26%
LMD	0.1585	0.1642	-	-
CEEMD	0.6262	0.3147	-	-
ICELMD and ARMA−KF	0.0487	0.0509	86.82%	98.29%

## 5. Conclusions

A novel bias drift suppression method is proposed based on two-fold ICELMD and ARMA-KF. First, ICELMD is applied to process the original signal, with abnormal noise removed by 2σ criterion. Then, ICELMD is applied to decompose the preprocessed signal, and the preprocessed signal is classified into noise, mixed and useful components by CMSE and PE over second ICELMD. After classification, the mixed components are modeled by ARMA, and denoised by KF. Experiments and comparisons reveal that the method has great performance on denoising, the angle random walk decreases about 97.98%, the zero bias instability decreases about 86.43%, the standard deviation decreases about 86.62%, and the variance decreases about 98.29%.However, the angle random walk of the signal denoised by LMD method decreases about 93.43%, and the zero bias instability decreases about 56.24%. The standard deviation of signal denoised by RNN-UKF is optimized by 67.22%, and the variance suppression is about 89.26%. From the results above, we find the combined ICELMD and ARMA-KF method has better denoising ability compared with LMD and CEEMD. Further, the method performs better than [25,28] on standard deviation and variance suppression.

## 6. Discussion

The LMD is first proposed to decompose the signal of electroencephalogram (EEG) and retrieve the specific features [24]. Then, there are many works applying LMD to analyze the signals of EEG or EMG as features extraction and tools of data analysis [29,30,31]. However, the performances of using pure LMD to denoise the signal of MEMS gyro is not magnificent. To our best knowledge, this is the first time that the Interoperated CELMD is proposed and applied twice in our paper. First, ICELMD is exploited in the preprocess to reduce the abnormal noise of MEMS gyro. Then, the ICELMD is applied again to classify and denoised together with ARMA-KF to ensure the super noise-suppressed effects. LMD is an adaptive way to analyze the signal, which is very suitable for non-stationary and non-linear signal processing. Therefore, the method could be extended to the sensors which detect random and non-stationary signals. However, the proposed model is effective for MEMS gyros, whose performances for other sensors may not as good as those for MEMS gyros. To achieve better performances, the proposed model needs, accordingly, modifications. The modifications include the choice of the specific classification and filtering methods for the sampled signals. Future works include developing novel denoising methods replacing ARMA-KF, which can indicate the mixed components more accurate and improve the performance. In addition, we may apply algorithms of Artificial Intelligence (AI) to train the model and realize real-time denoising.

## Figures and Tables

**Figure 1 micromachines-14-00109-f001:**
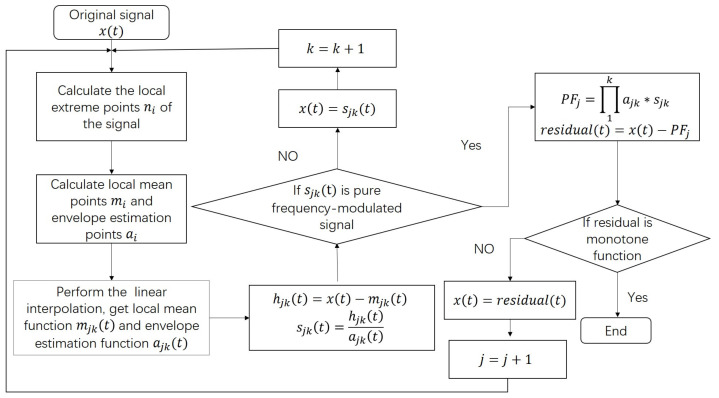
Flow chart of the proposed Interpolated LMD.

**Figure 2 micromachines-14-00109-f002:**
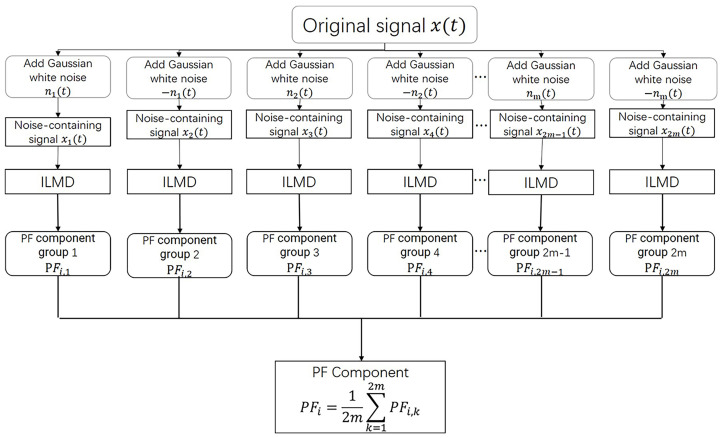
Flow chart of Interpolate CELMD.

**Figure 3 micromachines-14-00109-f003:**
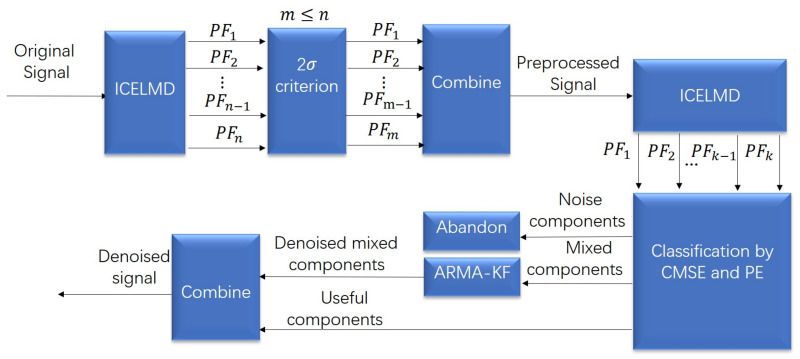
Overall process of the proposed ICELMD-ARMA model.

**Figure 4 micromachines-14-00109-f004:**
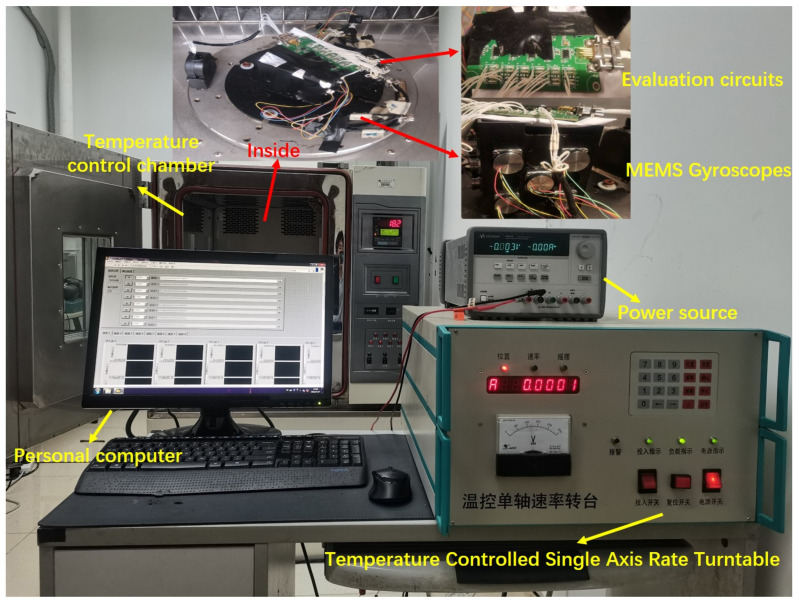
Experiment equipment.

**Figure 5 micromachines-14-00109-f005:**
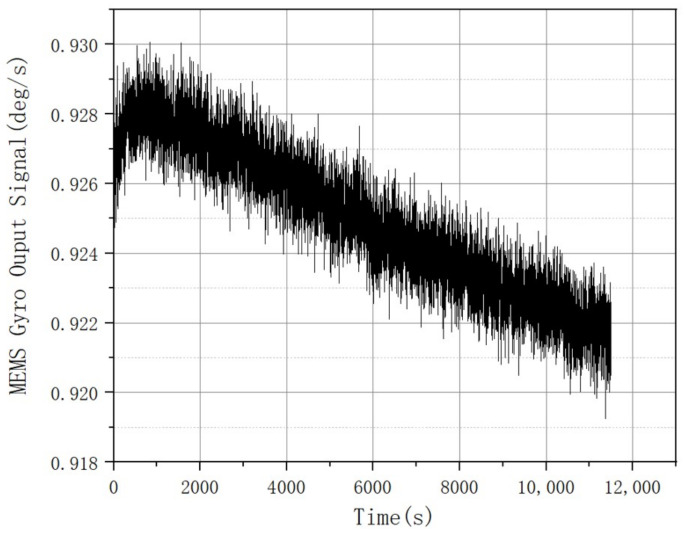
Original signal.

**Figure 6 micromachines-14-00109-f006:**
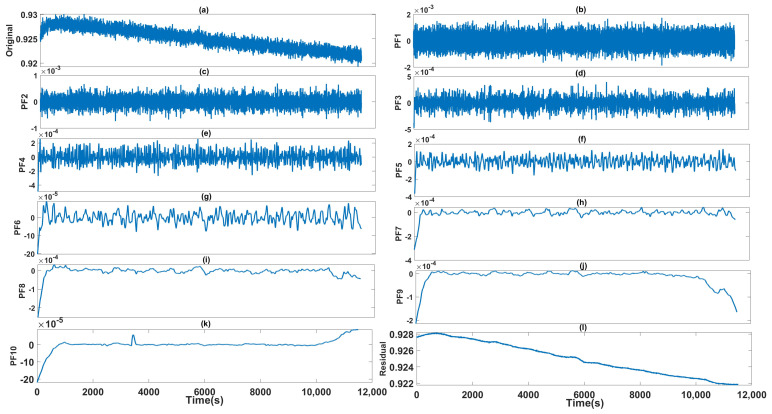
(**b**−**k**) are 10 PFs obtained by ICELMD, (**a**) is original signal and (**l**) is the residual signal.

**Figure 7 micromachines-14-00109-f007:**
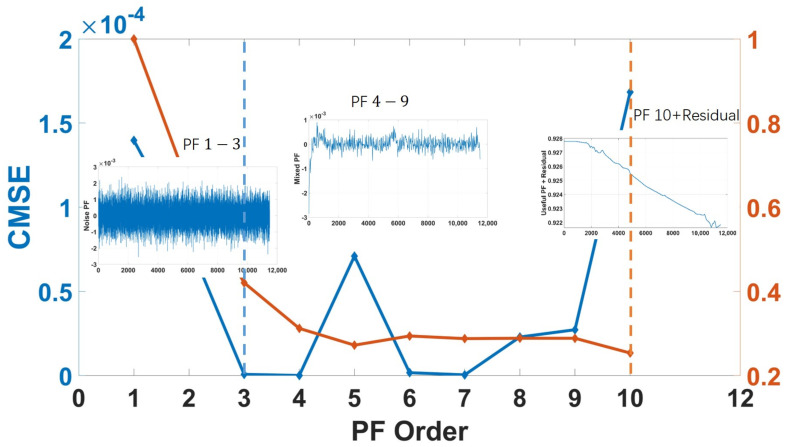
PF classification based on CMSE and PE.

**Figure 8 micromachines-14-00109-f008:**
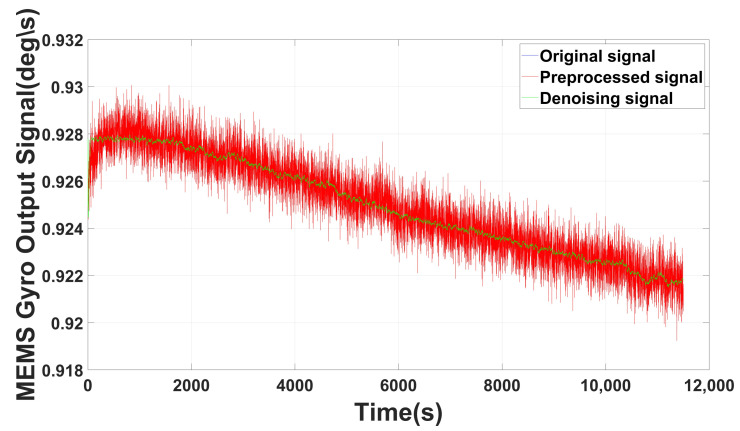
Denoising result of ICELMD (without abnormal signal).

**Figure 9 micromachines-14-00109-f009:**
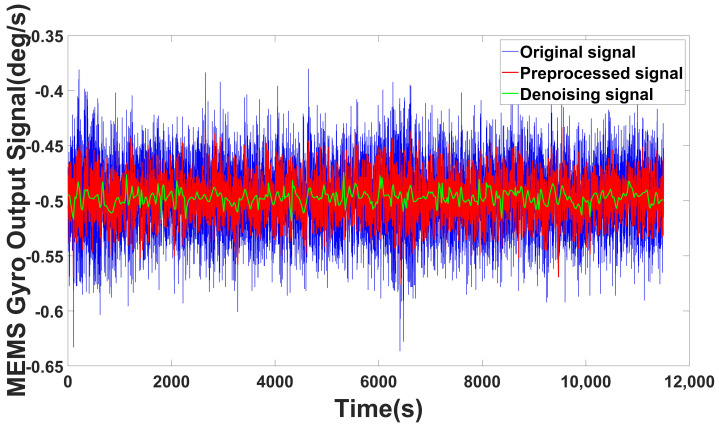
Denoising result of ICELMD (MPU6050 with abnormal signal).

**Figure 10 micromachines-14-00109-f010:**
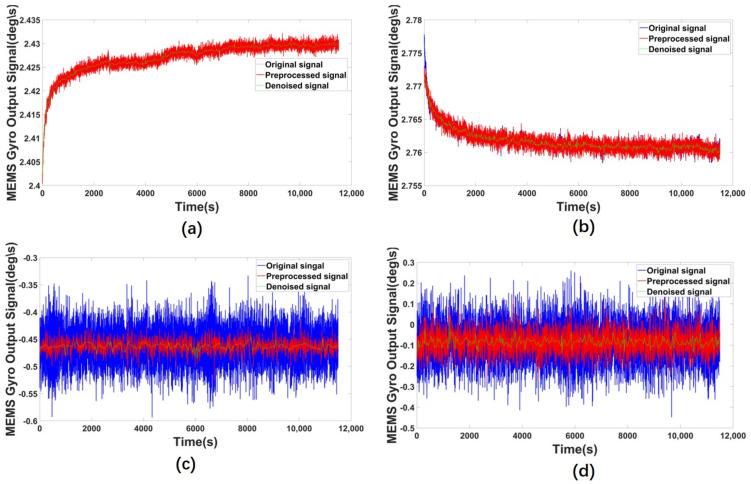
Denoising results of ICELMD, (**a**,**b**) are collected by different gyros, and (**c**,**d**) are collected by MPU6050.

**Figure 11 micromachines-14-00109-f011:**
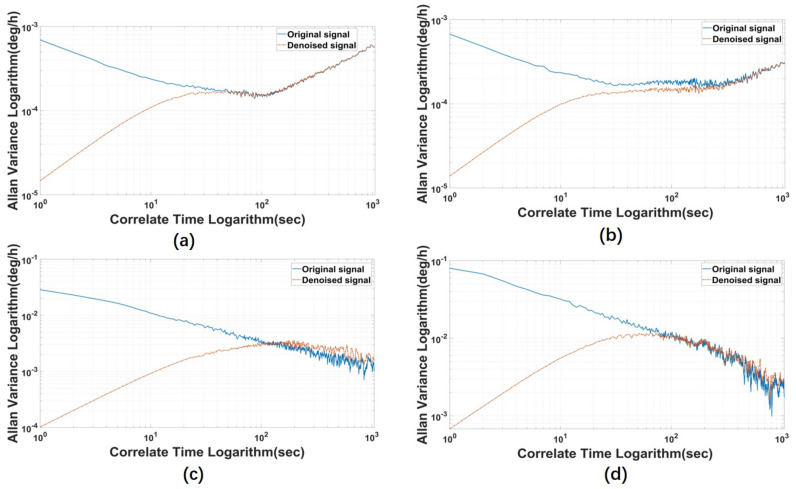
Allan variance of different denoising results of ICELMD, (**a**,**b**) are collected by different gyros, and (**c**,**d**) are collected by MPU6050.

**Figure 12 micromachines-14-00109-f012:**
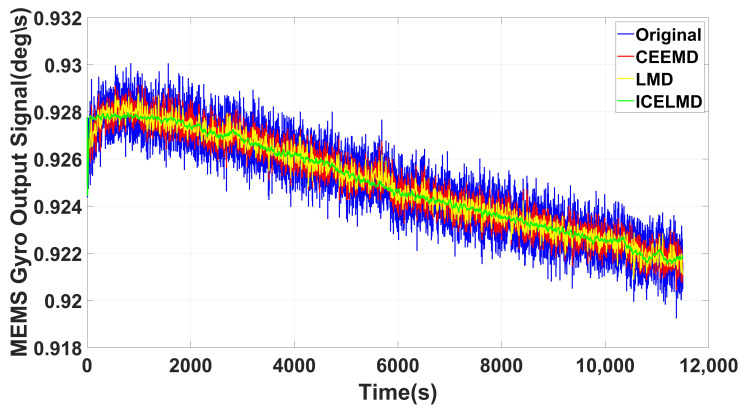
Denoising effects of different methods.

**Figure 13 micromachines-14-00109-f013:**
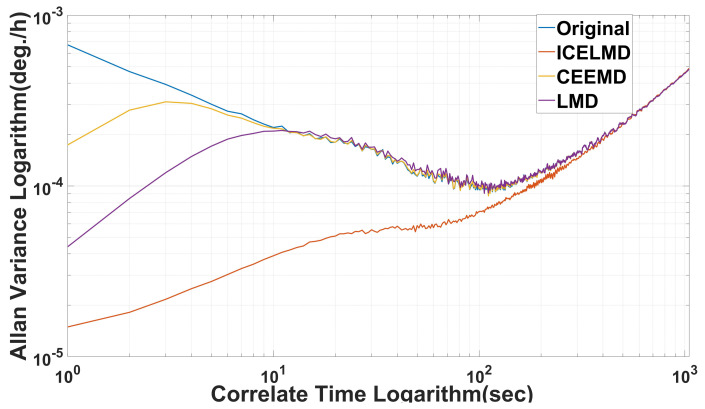
Allan variance of different denoising methods.

**Table 2 micromachines-14-00109-t002:** Time Series Model.

Model Name	Autocorrelation Function	Partial Autocorrelation Function
AR(p)	trailing	p-step censoring
MA(q)	q-step censoring	trailing
ARMA(p,q)	trailing	trailing

## Data Availability

Not applicable.

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
