# Peer review of "A Bias Drift Suppression Method Based on ICELMD and ARMA-KF for MEMS Gyros"

_micromachines, 2022, doi:10.3390/mi14010109_

Round 1

Reviewer 1 Report

See attachment

Reviewer 2 Report

The manuscript is reporting a new model for reducing the bias drift of the gyro sensors through a bias drift compensation primarily based on the Interpolated Complementary Ensemble Local Mean Decomposition (ICELMD) and ARMA-KF. While there have been good contributions, the manuscript will be publishable if the following issues are addressed.

1)     The abstract does not reflect the paper's main take-home message. Please clarify the novelty and clearly state the improvements made by your model.  

2)     I suggest that authors explain the novelties in separate bullet points in the introduction.

3)     Since many models are explained in the manuscript (in the introduction), please prepare a table/chart that reflects the advantages and disadvantages of each one compared to your model. This helps the reader to maintain interest when reading the introduction. There are many abbreviations. I suggest a diagram showing models in one glance.  

4)     How the CELMD and ARMA-KF are combined to reduce the environmental noises? How many noises were experimented on? Please explain in the experimental section.

5)     Is the model can be extended to other sensors? For instance, biosensors or any other sensors with a noisy response? If so, I suggest opening a discussion section at the end and talking about that.

6)     I don’t see in the report explain of how the authors performed the experiment, the devices used, and the connection of devices and components (maybe you put them in the appendix). It is poorly reported in section 4.

7)     On page two, what is the meaning of “… the 2σ criterion as preprocess”? Please clarify in the text.

8)     To reduce the length of the paper, it is suggested to bring the 2.1. and 2.2. in the appendix or a supplementary section, is their structure as it is necessary?

9)     The caption of Figure (1) needs more explanation.

10)  Following comment 5 may help better presentation by adding a flowchart instead of text for 2.1. and 2.2. sections.

11)  What do authors mean by “More white noise means more time to calculate over a large signal-to-noise ratio (SNR)”? is it mentioning the relationship between white noise and SNR?

12)  Section 2.3. needs to be clarified to understand; please rewrite it to make it more comprehensible.

13)  Figure 2 is important for readers; why are the captions too short? Please add more explanations.

14)  Same goes for Figure 3! please add a couple of sentences to explain the process in the flowchart.

15)  Figure 4 is poorly captioned. Please state the interconnections of each component, and … please notice that the coloring of the text is not attention-grabbing aesthetically!

16)  Again, the poor presentation of Figure 7. Please make the sure text inside the picture is readable, and the inset pictures are visible enough.

17)  Please try to reflect the noticeable contribution in Table 2 in both the Abstract and Conclusion.

18)  Please check the structure and English of the conclusion to make sure the novelty of the work is appropriately presented.

19)  Figure 6 is not labeled correctly, please label each diagram and explain shortly about the figure in the caption.

Please revisit these sentences and the manuscript, there are more incidents of this kind:

-There are about there kinds of methods to improve the performance of MEMS gyro

- it is suggested the numbering of bullet points be reorganized in the following sentences:

(1) Physical methods, these methods generally include optimizing the structure and improving the industrial technology, which are completed in the design and production stages. The commonly used methods are laser quality adjustment [7] , adding mass block [8] and circuit compensation; (2) Linear or single models

* read loses the intended meaning when following the text. Please bring the items 1,23 and then explain them.

- please make sure the structure of this sentence is current and the following sentences:

Recently, several adaptive signal analyzing methods, empirical mode decomposition (EMD) and Local Mean Decomposition (LMD), have been studied and applied for MEMS gyro [12–17]….

-        It essentially adaptively separates a nonlinear, non-stationary signal in the order of decreasing frequency according to the envelope characteristics… (sentence structure)

-        The more times the averaging, the smaller effect the white noise will have on the decomposition results…
